# Neurochemical Characterization of Neurons Expressing Estrogen Receptor β in the Hypothalamic Nuclei of Rats Using in Situ Hybridization and Immunofluorescence

**DOI:** 10.3390/ijms21010115

**Published:** 2019-12-23

**Authors:** Moeko Kanaya, Shimpei Higo, Hitoshi Ozawa

**Affiliations:** Department of Anatomy and Neurobiology, Graduate School of Medicine, Nippon Medical School, 1-1-5 Sendagi, Bunkyo-ku, Tokyo 113-8602, Japan; kanaya.moeko@twmu.ac.jp (M.K.); hozawa@nms.ac.jp (H.O.)

**Keywords:** estrogen receptor β, *Esr2*, RNAscope, kisspeptin, anteroventral periventricular nucleus, arcuate nucleus, vasopressin, oxytocin, corticotropin-releasing factor, paraventricular nucleus

## Abstract

Estrogens play an essential role in multiple physiological functions in the brain, including reproductive neuroendocrine, learning and memory, and anxiety-related behaviors. To determine these estrogen functions, many studies have tried to characterize neurons expressing estrogen receptors known as ERα and ERβ. However, the characteristics of ERβ-expressing neurons in the rat brain still remain poorly understood compared to that of ERα-expressing neurons. The main aim of this study is to determine the neurochemical characteristics of ERβ-expressing neurons in the rat hypothalamus using RNAscope *in situ* hybridization (ISH) combined with immunofluorescence. Strong *Esr2* signals were observed especially in the anteroventral periventricular nucleus (AVPV), bed nucleus of stria terminalis, hypothalamic paraventricular nucleus (PVN), supraoptic nucleus, and medial amygdala, as previously reported. RNAscope ISH with immunofluorescence revealed that more than half of kisspeptin neurons in female AVPV expressed *Esr2*, whereas few kisspeptin neurons were found to co-express *Esr2* in the arcuate nucleus. In the PVN, we observed a high ratio of *Esr2* co-expression in arginine-vasopressin neurons and a low ratio in oxytocin and corticotropin-releasing factor neurons. The detailed neurochemical characteristics of ERβ-expressing neurons identified in the current study can be very essential to understand the estrogen signaling via ERβ.

## 1. Introduction

Estrogens exert actions through nuclear estrogen receptor α (ERα) and β (ERβ). In adulthood, ERα has been predominantly linked to reproductive neuroendocrine and sexual behaviors [1,2], whereas ERβ has been associated with many non-reproductive neurophysiological functions such as learning and memory [3,4], anxiety-related behaviors [5,6], and the stress response of the hypothalamic-pituitary-adrenal (HPA) axis [7]. To fully understand the involvement of estrogen receptors (ERs) in these functions, it is important to identify the neuroanatomical and neurochemical organization of neurons expressing ERs. The distributions of both subtypes in the brain and their neurochemical phenotypes using *in situ* hybridization (ISH), immunohistochemistry, and PCR have been investigated in current literature [8,9,10] and many reports have successfully determined various characteristics of ERα-expressing neuronal populations [11,12]. However, and unlike ERα, ERβ’s respective protein distribution has been fairly controversial in some immunohistochemistry-based studies because of the absence of specific antibodies [13]. Studies using ISH for *Esr2* mRNA (which encodes ERβ), are also limited partially because of methodological complications in terms of visualizing low-expression *Esr2* gene in the brain. Furthermore, the reduction of immunoreactivity following ISH is problematic in further neurochemical characterization of ISH-based studies.

The majority of previous reports has consistently shown that, unlike ER*a*, ERβ is densely expressed in the paraventricular nucleus of the hypothalamus (PVN) and supraoptic nucleus (SON). Both nuclei express arginine vasopressin (AVP) and oxytocin (OXT) neurons, whereas the PVN additionally expresses the corticotropin-releasing factor (CRF) [14,15]. These heterogenous populations of neurons in the PVN have been shown to control a wide array of autonomic and neuroendocrine functions. In fact, some studies have indicated an involvement of ERβ-mediated regulation in these physiological functions, such as the regulation of the HPA axis [15]. However, little is reported regarding the detailed characteristics of these ERβ-expressing neurons such as the co-expression ratio in AVP, OXT, and CRF neurons, the regional difference in the co-expression ratio depending on rostro-caudal of the PVN, and the respective sexual dimorphism.

Furthermore, the anteroventral periventricular nucleus (AVPV) and arcuate nucleus (ARC) have been reported to express ERβ [9,16]. In both nuclei, estrogens play a central role in the regulation of the reproductive axis via kisspeptin neurons, key regulators of gonadotropin releasing hormone secretion. Most kisspeptin neurons in female AVPV express ERα [17,18]. On the other hand, kisspeptin neurons in the ARC co-express ERα in females [19], and also in males [20]. Therefore, estrogens are thought to be involved in reproduction by a process that is mediated by ERα. However, the co-expression ratio of *Esr2* and kisspeptin is still unclear.

In this study, we aim to provide accurate and detailed data regarding the neurochemical characteristics of neurons expressing *Esr2* mRNA in the hypothalamus of female and male rats using RNAscope ISH for *Esr2* mRNA combined with immunofluorescence against kisspeptin, AVP, OXT, and CRF. We used RNAscope ISH because of its high-sensitivity and low-background, as well as its minimal immunoreactivity impairment following ISH [21]. First, we performed experiments using two methodologies to validate the RNAscope signals for *Esr2* mRNA: comparison with conventional ISH with digoxigenin (DIG)-RNA probes and qPCR-based verification. Following confirmation that the RNAscope results were in good agreement with the results obtained by these two methodologies, we investigated the co-expression of *Esr2* mRNA and neuropeptides using RNAscope ISH combined with immunofluorescence.

## 2. Results

### 2.1. Validation of RNAscope Signals for Esr2

To confirm the validity and specificity of RNAscope signals for *Esr2*, we compared the results of RNAscope ISH using two methodologies: conventional DIG-labeled RNA ISH and qPCR. Using single RNAscope ISH, *Esr2* signals (red particles) were densely observed in the AVPV, bed nucleus stria terminalis (BNST), SON, PVN, and medial amygdala (MeA) (Figure 1A,B). Likewise, conventional ISH using DIG-RNA probes revealed *Esr2* mRNA positive cells in these nuclei (Figure 1C), which were relatively obscure signals compared to the ones detected by the RNAscope. Furthermore, we identified *Esr2* signals in the organum vasculosum laminae terminalis (OVLT), the ventrolateral hypothalamic tract (vlh), and in the ventrolateral parts of ventromedial hypothalamus (VMHvl) (Appendix A), even though the signals in these regions were not significantly strong. We confirmed RNAscope *Esr2* signals in the positive control tissues of ovarian granulosa cells and prostate epithelial cells (Appendix A). On the other hand, the absence of signals was confirmed in multiple negative controls including RNase-treated sections, omission of *Esr2* probe, and probes for *DapB* gene encoding bacterial dihydrodipicolinate reductase, which is not expressed in any eukaryote, and DIG-labeled sense probes (Appendix A).

Consequently, we assessed the *Esr2* mRNA in the laser-microdissection (LMD)-isolated tissues of the AVPV, SON, PVN, suprachiasmatic nucleus (SCN), and corpus striatum (CP) to validate RNAscope signals with qPCR (Figure 1D). SCN and CP were chosen as representative nuclei where no signal of *Esr2* were found by RNAscope ISH (Appendix A). We determined that *Esr2* mRNA was expressed in the AVPV, SON, and PVN, whereas *Esr2* mRNA was expressed in the SCN and CP was not detectable (Figure 1E). This is a finding in good agreement with the RNAscope ISH stained images.

### 2.2. Co-Expression of Esr2 and Neuropeptides

#### 2.2.1. Co-Expression of *Esr2* in Kisspeptin Neurons in the AVPV and ARC

Figure 2A,B show kisspeptin neurons co-expressing *Esr2* in the AVPV of female and ARC of both sexes, respectively. RNAscope ISH and immunofluorescence revealed that approximately 70% of kisspeptin-ir cells in the AVPV co-expressed *Esr2* mRNA (Figure 2C). Cells with more than three signal particles of *Esr2* were considered as “co-expressed.” Conversely, the low percentage of co-expression of *Esr2* mRNA in kisspeptin neurons in the ARC was found in a sexually dimorphic manner with females having a higher co-expression ratio than males (25.1% in females and 5.5% in males, *p* < 0.05, Figure 2D). Furthermore, the number of kisspeptin-ir cells was found to be greater than the one found in male ARC (180.0 ± 29.0 cells in females and 41.5 ± 6.20 cells in males, *p* < 0.05). The composition of cells, which have more than five particles/cell within *Esr2*-expressing cells was 87% AVPV kisspeptin, and 45% in ARC kisspeptin of females (Appendix A). Note that we did not perform analysis in male AVPV because males have few kisspeptin neurons in the AVPV.

#### 2.2.2. Co-Expression of *Esr2* in the AVP, OXT, and CRF Neurons in the PVN

Figure 3 shows the co-expression of *Esr2* and AVP in the PVN at different rostro-caudal levels. Two-way ANOVA revealed that the AVP containing *Esr2* percentage did not vary across the rostro-caudal axis of the PVN. Moreover, there was no sex difference nor did an interaction between these two main factors occur. Both female and male rats showed that the majority of AVP neurons co-expressed *Esr2* in the PVN (Figure 3C). Co-expression percentages of *Esr2* in AVP neurons throughout the PVN in total were 54.9% and 50.3% in females and males, respectively (not significant). Within these *Esr2*-expressing AVP neurons in the PVN, 74% and 80% of neurons showed ≥5 signal particles/cell in females and males, respectively (Appendix A).

Two-way ANOVA underlined that rostro-caudal levels in the PVN had a significant effect on the co-expression of *Esr2* and OXT (*p* < 0.05) as opposed to sex and the interaction between the two main factors. The *Esr2* co-expression rate in OXT neurons in the PVN were 12.5% and 23.8% in females and males, respectively (not significant). In the PVN at the caudal level (from bregma −2.04 to −2.16 mm), the co-expression ratio was 0% in both sexes because OXT neurons were not detected (Figure 4A,B). Within these *Esr2*-expressing OXT neurons, 55% and 51% of neurons showed ≥5 signal particles/cell in females and males, respectively (Appendix A).

Additionally, and despite the fact that the number of co-expressed cells was low, two-way ANOVA indicated a significant effect of rostro-caudal levels of the PVN on the co-expression ratio of *Esr2* in the CRF neurons (*p* < 0.05). The main effect of sex and the interaction between rostro-caudal levels of the PVN and sex on the co-expression ratio was not statistically significant. Total co-expression ratio of *Esr2* in CRF neurons in the PVN was 6.5% in females and 12.2% in males (Figure 4C,D).

#### 2.2.3. Col-Expression of *Esr2*/AVP and *Esr2*/OXT in the SON

Co-expressions of *Esr2*/AVP and *Esr2*/OXT in the SON are shown in Figure 5A,B. The percent of AVP neurons co-expressing *Esr2* in the SON was 54.0% in females and 60.9% in males (Figure 5C). In contrast, and the percent of OXT co-expressing *Esr2* in the SON was low (15.4% in females and 12.8% in males, Figure 5D). Additionally, the majority of *Esr2*-expressing AVP neurons in the SON showed ≥5 particles/cell (85% in females and 79% in males), whereas the percentage of cells with ≥5 particles/cell in *Esr2*-expressing OXT neurons was low (36% in females and 37% in males) (Appendix A).

## 3. Discussion

### 3.1. Effectiveness of RNAscope in Detecting Esr2 mRNA in the Rat Brain

In this study, we used RNAscope, which is a novel ISH technology, to provide detailed data of *Esr2* mRNA co-expressing neuropeptides in the rat hypothalamus. Validation analysis by comparison of RNAscope *Esr2* with conventional ISH and qPCR revealed that the RNAscope allows for the accurate detection of mRNA signals for *Esr2*. RNAscope generated clear and visible signals that led to a higher S/N ratio compared to the one acquired by the conventional ISH with DIG-labeled probes. Moreover, RNAscope is extremely advantageous for estimating the mRNA levels of target genes in individual cells because the number of signal-dots reflect the expression levels of target genes. Therefore, these profound merits can facilitate the precise and distinct quantification of the ISH signals. In addition to the strong signals in the nuclei mentioned above, we observed moderate RNAscope signals in several brain regions including OVLT and VMHvl, as opposed to the weak obscure signals or no signals detected by conventional ISH in these regions, which suggests that sensitivity of RNAscope for *Esr2* was qualitatively better than that of conventional ISH. RNAscope can efficiently be used to detect target genes, especially when these genes are expressed at very low levels in tissues.

Several splice variants of ERβ mRNA have been reported in the brain [22] such as ER-β2 variant (54 bp insertion between exons 5 and 6), and δ3 variants (deletion of the third exon). It is suggested that these isoforms play roles in differential regulation of estrogen among brain regions [23] as a result of the functional diversity of these isoforms in many properties including the transcriptional activation and ligand-binding affinity. In this study, both the DIG-labeled probe and RNAscope probe can detect these variants as well as wild-type *Esr2*, while it is difficult to determine the expression profile of these isoforms using these probes because of its pan-specificity to multiple isoforms. Further characterization of the isoform profile in each *Esr2*-expressing neuronal population is needed to better understand the role ERβ functions in the multiple brain areas.

### 3.2. Kisspeptin Neurons Colocalizes with Esr2 in a Region-Specific and Sex-Specific Manner

In this study, we determined that approximately 70% of kisspeptin neurons in female AVPV co-expressed *Esr2*, whereas less than 30% kisspeptin neurons in the ARC co-expressed *Esr2*. Kisspeptin neurons in the AVPV and ARC are responsible for the estrogen-mediated generation of the surge and pulse modes of gonadotropin releasing hormone/luteinizing hormone secretion, respectively [24,25]. In female rodents, kisspeptin in the AVPV and ARC is regulated by estrogens in an opposite manner. Kisspeptin is up-regulated in the AVPV by estrogens, whereas kisspeptin is down-regulated in the ARC. These opposing effects promoted by estrogens between the two regions may be partially explained by regional differences observed in this study in ERβ expression. Most of kisspeptin neurons in females, in both the AVPV and ARC, co-express ERα mRNA in rodents [19]. However, kisspeptin neurons in mice lacking functional ERα fail to respond to estrogens in either the AVPV or ARC [19]. Thus, the importance of ERα for mediating the inhibitory and stimulatory effects of estrogens on kisspeptin neurons is self-evident in females. Nevertheless, high co-expression rates of *Esr2* in the kisspeptin neuron in the AVPV indicate that kisspeptin in the AVPV is directly regulated by not only ERα but also ERβ. In support of this argument, it has been reported that AVPV-specific ERβ downregulation using siRNA results in abnormal estrous cycles, and, consequently, ERβ knockout mice tend to exhibit decreased litter size [16,26]. Certain studies have also reported that a smaller fraction of kisspeptin neurons in the AVPV expresses *Esr2* mRNA in females (21% in rats and 30.5% in mice) [11,19]. This inconsistency may come as a result of the different steroid condition (intact rats at metestrus versus sex steroid-replaced gonadectomized rats) and/or of the different detecting target molecule used (protein versus mRNA). On the other hand, in the ARC, almost all ARC kisspeptin neurons co-express ERα in both sexes [19,20]. Mice with ERα knocked down in ARC kisspeptin neurons exhibit disrupted cyclicity [27]. Therefore, and combined with our results, estrogen action on reproduction through ARC kisspeptin neurons is exerted primarily via ERα rather than ERβ. In this study, kisspeptin expression showed a clear sex-difference with lower expression in males than females in the ARC, which is consistent with our previous study [28]. It is suggested that not only estrogen but also testosterone play a role in this sexual dimorphism in kisspeptin. The perinatal testosterone surge persistently suppresses kisspeptin after aromatization to estradiol in the brain [29,30,31], and high circulating testosterone also suppresses kisspeptin expression in adulthood [30]. Co-expression analysis revealed distinct sex differences in *Esr2* co-expression in kisspeptin neurons in the ARC. This result may reflect the sex difference in the total number of kisspeptin cells in the ARC (180.0 ± 29.0 cells in females vs. 41.5 ± 6.20 cells in males), as a consequence of the difference in intrinsic sex steroid milieu between sexes.

### 3.3. Majority of AVP Neurons Co-Express Esr2 in the PVN and SON in Both Sexes

In this study, we identified the presence of *Esr2* mRNA in the majority of AVP neurons in the PVN and SON. Moreover, the percentage of cells with ≥5 *Esr2* signals was relatively higher in the AVP neurons compared with OXT neurons in the PVN and SON. Current literature is inconsistent when it comes to evaluating the neurochemical characteristics of ERβ-expressing neurons in the PVN. The significant co-expression of *Esr2* in AVP neurons found in this study is consistent with findings of previous studies [14,32], whereas the predominant colocalization of ERβ-ir in OXT neurons has also been reported [33,34] in the PVN. Hence, inconsistencies may be the result of a different steroid status. Ovariectomized female rats were used in the previous reports by Simonian and Herbison and Alves et al., whereas, in this study, as well as in Suzuki and Handa, females were gonadally intact and we also examined intact males. Although gonadectomized animals are a widely used model for examining estrogen actions, the intact model used in this study reflects the physiological conditions. An additional possibility may be the lack of specificity of antibodies for ERβ used in previous studies [35]. Regarding the PVN of mice, Oyola et al. reported a relatively low ERβ/AVP co-expression with a sex difference (45% in females versus 11% in males in the caudal PVN) [36,37] using transgenic mice model expressing ERβ identified by enhanced green fluorescent protein. This group also reported a high co-expression of ERβ in OXT neurons (>80% in both sexes) in the PVN. These findings suggest there exists a species difference in neurochemical characteristics of *Esr2*-expressing neurons between rats and mice. In the SON, our present result is consistent with previous studies in which the co-localization of AVP neurons with ERβ-ir cells is high (72% [32] and 88% [14]). Significant overlap between *Esr2* mRNA signals and AVP-ir cells in both the PVN and SON indicates that AVP neurons could be regulated by estrogens through an ERβ-dependent pathway. Likewise, estrogens have been shown to directly regulate AVP promoter activity through an ERβ in rats [38]. A well-documented physiological function of AVP neurons in both the PVN and SON is the maintenance of water balance, which is influenced by estrogens [39,40]. The neurons responsible for the maintenance of water homeostasis are the magnocellular neurosecretory cells located in the PVN and SON that projects its axons toward the posterior pituitary and is, subsequently, released into the circulation. Although there was no statistical significance in this study, we pinpointed a tendency of more cells co-expressed of *Esr2* mRNA in AVP neurons in the caudal levels (−1.44 to −2.16 mm from bregma) in the PVN, in which many magnocellular cells are located. These results indicate that ERβ-expressing AVP neurons could be involved in the water homeostasis via the secretion of AVP into systemic circulation. AVP in the PVN is also considered to participate in stress responses by modulating the HPA axis [41,42]. Although sex differences in HPA-axis reactivity have been reported, with females demonstrating a more robust stress response than males [43], we observed no sex difference in the co-expression of *Esr2* mRNA and in AVP neurons. Thus, the sex difference in HPA-axis activity may not be influenced by the level of ERβ/AVP. Lund et al. (2005) reported that ERα and ERβ have opposing effects on the HPA-axis activation [5]. However, involvement of direct ERα-signaling in the AVP is unlikely because the PVN contains very little ERα [44]. Hence, involvement of estrogens in the sex difference of HPA-axis activity needs to be investigated while considering multiple possibilities such as estrogen action on the stress-responsive neuronal populations projecting to the PVN, and context-dependent change in ERα/ERβ expression in the PVN. Additionally, a recent study disclosed that galanin-expressing neurons in the medial preoptic area directly receive the inputs from the AVP neurons in the PVN and SON without a clear sex difference [45]. The galanin-expressing neurons in the medial preoptic area are associated with parental behaviors, and, therefore, the estrogen action via ERβ on AVP neurons in the PVN partially results in an effect on the mechanism of parental behaviors in both sexes.

### 3.4. OXT Neurons Co-Expressing Esr2 were Present at Low Levels in the PVN and SON in Both Sexes

In the present study, we found a low ratio of OXT neurons co-expressing *Esr2* mRNA in the PVN and SON. The inter-PVN distribution of OXT-expressing neurons was slightly rostrally-biased compared to the AVP populations, with a maximum cell count at around −1.32 mm from bregma (data not shown), where parvicellular cells are mainly located. Apparently, this result is different from several previous studies in which high percentages of colocalized ERβ/OXT were found [8,14,32]. However, these studies show high co-expression in the parvocellular parts, but not in the magnocellular parts of the PVN. In fact, our study considers the total number of the co-expression ratio through the PVN. Furthermore, and although our results show a relatively low co-expression of ERβ in the OXT neurons in a physiological condition, many studies have suggested a strong relationship between ERβ and OXT. More specifically, treatment of estrogen and ERβ selective agonist enhances OXT mRNA levels in the rat hypothalamus [46,47], and activate the OXT promoter [48]. Because ERα levels in the PVN are almost negligible [32], estrogens may exert their action on OXT secretion by activating the ERβ-expressing subpopulation in the PVN OXT neurons. When it comes to stress responses, OXT administration into the PVN inhibits the HPA axis in male and female rats [49] and ERβ and OXT cooperate to reduce anxiety-like behaviors and HPA-axis responses [50]. Therefore, estrogens have a direct effect on the expression of OXT through ERβ in the parvocellular parts of the PVN and may interact to regulate HPA axis, even at low co-expressed levels. Previous studies have underlined the low co-expression of ERβ in the magnocellular OXT neurons in the PVN [32,33], which is a finding that is consistent with the low expression in the caudal subpopulation of the PVN OXT neurons found in this study. In addition, in the SON, we observed that the co-localization of OXT neurons with ERβ-ir cells is relatively lower than AVP neurons as a previous study [14]. However, estrogens are known to regulate milk ejection and parturition [51] through the activation of the magnocellular OXT neurosecretory cells in the PVN and SON. Hence, further analysis of ER expression in pregnant and lactating animals is necessary. In addition to its neuroendocrine function, OXT neurons in the PVN play roles in modulating a wide variety of brain functions through the direct axonal projections to multiple brain areas such as cortex and central amygdala. The former neuronal pathway seems to be implicated in maternal behavior [52], and the latter one is related to a fear response [53]. In addition, parvocellular neurons in the PVN project to sympathetic and parasympathetic centers in the brainstem and spinal cord [54]. It has been proposed to the involvement of PVN OXT neurons in the modulation of sympathetic and parasympathetic outflows to the spinal cord [55]. Further analysis of ERβ expression in each OXT neuronal populations depending on its projection site is necessary to fully elucidate the ERβ functions.

### 3.5. CRF Co-Expressing Esr2 mRNA Neurons Are almost Absent in the PVN

We found a very small percentage of *Esr2* mRNA/CRF co-expressed neurons in the PVN, especially in the parvocellular cells, which is consistent with a previous study [32]. However, mRNA and protein levels of CRF in the PVN are enhanced by estrogens [56,57], which implies an interaction between ERβ and CRF. On the other hand, and when it comes to physiological functions, CRF is a key player in controlling the HPA axis [58] and AVP also works as a secretagogue for the adrenocorticotropic hormone to potentiate CRF activity at the pituitary. These neurons are located in the parvocellular parts of the PVN, which terminate in the median eminence and can effectively coordinate secretion of CRF and AVP [59]. Considering the low percentage of *Esr2*/CRF and *Esr2*/AVP in the parvocellular parts in the current study, further experiments are required to determine the indirect action of estrogens on the HPA through AVP and CRF neurons expressing *Esr2*. In contrast to this result, in mice, the number of CRF neurons co-expressing ERβ is around 30% to 40% in the middle and caudal levels of the PVN of both sexes [36]. There may be a species difference in the co-expression pattern as well as within the AVP and OXT neurons.

## 4. Materials and Methods

### 4.1. Animals

Adult male and female Wistar–Imamichi rats for breeding were purchased from the Institute for Animal Reproduction (Ibaraki, Japan). Offspring derived from mating in our facility were housed with dams in the same cages until weaning at three weeks of age. After 8–10 weeks of age, female rats were confirmed to exhibit at least two standard four-day estrous cycles by daily vaginal smears. All animals were bred and housed in a room with a controlled temperature (24 ± 2 °C) and a 14-h light/10-h dark cycle (lights on 06:00 h). Standard diet and tap water were available ad libitum. All animals were euthanized under deep anesthesia by intraperitoneal injection of sodium pentobarbital (64.8 mg/kg body weight) and butorphanol (2.5 mg/kg body weight, Vetorphale, Meiji Seika Pharma, Tokyo, Japan). The time periods for euthanasia ranged from 15:00 to 18:00 h. All animal experimental procedures were approved by the Committee on the Care and Use of Experimental Animals of Nippon Medical School and were conducted in compliance with Guidance on Animal Bioethics (based on the guidelines for humane treatment of experimental animals issued by the US National Institutes of Health) of Nippon Medical School (permit number 27-173; approved 22 June 2015).

### 4.2. Experimental Design

#### 4.2.1. Experiment 1: Validation of *Esr2* mRNA Signals in RNAscope ISH

To validate *Esr2* mRNA signals by RNAscope ISH, we compared the *Esr2* RNAscope signals with the distribution of *Esr2* mRNA by conventional ISH. We confirmed the presence of RNAscope signals for *Esr2* in several anatomical regions in which the DIG-labeled ISH showed positive signals: the AVPV, BNST, PVN, SON, and MeA. Consequently, we verified the validity of RNAscope by confirming that these results are consistent with the ones derived by a qPCR analysis. In this analysis, we used laser-micro-dissected samples from three RNAscope-signal rich brain regions, and two regions where the RNAscope signal was absent.

#### 4.2.2. Experiment 2: Characterization of *Esr2* mRNA-Expressing Neurons by a Combination of RNAscope ISH and Immunofluorescence

To determine co-expression of *Esr2* mRNA-expressing cells with each neuropeptide, we performed RNAscope ISH for *Esr2* followed by immunofluorescence against kisspeptin, AVP, OXT, and CRF. We then analyzed *Esr2* co-expression in kisspeptin neurons in the AVPV and ARC, in AVP neurons, OXT neurons in the PVN and SON, and CRF neurons in the PVN.

### 4.3. General Procedure

#### 4.3.1. Tissue Preparation

Female rats at the metestrus stage and male rats were transcardially perfused with 0.05 M phosphate-buffered saline (PBS, pH, 7.4) followed by 4% paraformaldehyde in 0.05 M phosphate buffer (PB, pH, 7.4) from 15:00 to 18:00 h. Their brains were post-fixed in the same fixative at 4 °C overnight and, subsequently, equilibrated to 20% sucrose in 0.05 M PB at 4 °C for two–three days until they sank. Four series of brain coronal sections were cut at a thickness of 16 µm for single RNAscope ISH and for a combination of RNAscope and immunofluorescence, and cut at a thickness of 40 µm for conventional ISH with DIG-RNA probe using a cryostat (CM3050; Leica, Wetzlar, Germany). We used 40-µm thick sections for DIG-ISH in accordance with our previous study [54], in which we optimized the ISH procedure for detection of low-expression targets. Each series of sections was stored separately in cryoprotectant at −25 °C and then used for single RNAscope ISH, a combination of RNAscope and immunofluorescence, and conventional ISH. For qPCR analysis, female rats at the metestrus stage were deeply anesthetized and decapitated. Fresh brains were immediately frozen in hexane that was chilled to −80 °C and stored at −80 °C until further processing.

Colchicine treatment can enhance immunoreactivity in the somata of kisspeptin [60], and CRF [61] through inhibition of neuronal transport. Thus, we administered colchicine to female rats at proestrus between 15:00 h and 18:00 h, when kisspeptin expressing cells reach peak level during the estrous cycle, to analyze the co-expression between *Esr2* and kisspeptin in the AVPV [11]. Since males have few kisspeptin neurons in the AVPV [62], we did not investigate this co-expression in male rats. Female rats at metestrus and male rats underwent the colchicine administration for analysis of co-expression between *Esr2* and kisspeptin in the ARC and between *Esr2* and CRF in the PVN. Colchicine (50 µg/10 µL saline, Wako Chemicals, Tokyo, Japan) was administered intracerebroventricularly under isoflurane anesthesia (2% (*v*/*v*)) using a stereotaxic instrument (SR-5R, Narishige, Tokyo, Japan). The tip of a Hamilton syringe (10 µL) was lowered to a point 0.8 mm caudal to bregma, 3.6 mm below to dura, and 1.4 mm lateral to the midline. Following twenty-four hours colchicine injection, all rats were scarified by perfusion.

#### 4.3.2. Single RNAscope ISH for *Esr2* mRNA

Single RNA ISH was performed using the 2.5 HD Reagent Kit-RED (Cat. No. 322350, Advanced Cell Diagnostics (ACD), Hayward, CA, USA), according to the manufacturer’s protocol. After washing in PB, sections were mounted on Super frost slide glass (Matsunami Glass Ind., Osaka, Japan) and were dried on a hotplate at 60 °C for 30 min. The slides were incubated in a boiling (98–103 °C) solution of 1× Target Retrieval for 5 min. The slides were washed in PBS and were, consequently, dehydrated in graded EtOH (50%, 75%, and 100%) for 3 min each and dried with cool air using an ImmEdge Pen (Cat. No. H-4000, Vector) prior to allowing the development of a hydrophobic barrier around the tissue. Samples were incubated in proteinase solution at 40 °C for 30 min before washing in PBS, and then incubated with *Esr2* probe at 40 °C for 2 h. All remaining washing steps were performed using 1× wash buffer in the kit, unless otherwise stated. Subsequently, amplification steps using Amp1–6 in the kit were performed. ISH signals for *Esr2* were detected with a Fast Red substrate reaction for 10 min at room temperature (RT). Sections were then counterstained with hematoxylin. After washing in PB, samples were completely dried on a hot plate set at 60 °C and cover-slipped with EcoMount (ACD). The *Esr2* probe was designed by the manufacturer (NM_012754.1, bp897–1765, Cat. No. 317161, ACD). We prepared three types of negative controls to investigate whether the *Esr2* probe manufactured by ACD binds non-specifically to tissues. First, we confirmed the absence of RNAscope signals in RNase-treated sections (40 mg/mL at 37 °C for 30 min). We also confirmed the absence of non-specific signals in the sections treated with *DapB* (Cat. No. 310043, ACD), which is an ACD negative control probe because this bacterial gene is not expressed in any eukaryote. Furthermore, there were no signals found in the sections where the *Esr2* probe was omitted. Lastly, we detected *Esr2* mRNA by using the RNAscope in the ovary and prostate as a tissue positive control because these tissues are well-known to express ERβ [63,64,65].

#### 4.3.3. Conventional ISH for *Esr2* mRNA Using a DIG-Labeled Probe

Free-floating brain sections were washed with diethylpyrocarbonate (DEPC)-treated PBS and then incubated with 1 µg/mL proteinase K (Takara Bio, Otsu, Japan) in 10 mM Tris buffer (pH 7.4) and 10 mM ethylenediaminetetraacetic acid (EDTA, pH 8.0) at 37 °C for 25 min. After washing in DEPC-PBS, brain sections were incubated with 0.25% (*v*/*v*) acetic anhydride in 0.1 M triethanolamine (pH 8.0) for 10 min at RT. Brain sections were washed twice in DEPC-PBS and then incubated with prehybridization buffer (1× hybridization solution (Sigma–Aldrich, St Louis, MO, USA) containing 50% (*v*/*v*) formamide and 10% (*w*/*v*) dextran sulfate) at 60 °C for 30 min. Brain sections were hybridized with DIG-labeled antisense or the sense RNA probe for *Esr2* at 60 °C for 16 h. DIG-labeled antisense and sense RNA probe was synthesized from template cDNA of *Esr2* (position 1275–2240, GenBank accession No. NM_012754.1) using a DIG-RNA labeling kit (Roche Diagnostics, Mannheim, Germany). Consequently, brain sections were washed with 4x saline sodium citrate (SSC) containing 50% (*v*/*v*) formamide at 60 °C for 20 min, and then 2× SSC at 60 °C for 15 min. The sections were incubated with RNaseA (Roche Diagnostics, 20 µg/mL in 10 mM Tris-HCl (pH 8.0), 1 mM EDTA, and 500 nM NaCl) at 37 °C for 20 min, which is followed by washing with 2× SSC and 1× SSC at 60 °C for 20 min each time. After blocking with 1% (*w*/*v*) bovine serum albumin (Roche Diagnostics) at RT for 30 min, the sections reacted with an alkaline phosphatase conjugated anti-DIG antibody (1:1000, Roche Diagnostics) at 37 °C for 2 h. For visualization purposes, sections were treated with a chromogen solution containing nitro-blue tetrazolium and 5-bromo-4-chloro-3-indolyl phosphate (Roche Diagnostics) overnight at 4 °C. Thereafter, the sections were mounted on gelatin-coated slides and cover-slipped with 90% (*v*/*v*) glycerol in PBS.

#### 4.3.4. Analysis of *Esr2* mRNA Expression in the Nuclei Isolated by Laser Microdissection

For qPCR analysis, we isolated nuclei from brain sections using a LMD system (Leica LMD 7000; Leica). We dissected the AVPV, PVN, and SON as RNAscope-signal-rich nuclei and SCN and CP as RNAscope-signal-negative nuclei in accordance with the procedure reported previously [66], with slight modifications. Frozen brains were cut coronally at a thickness of 30 µm on a cryostat. Brain sections containing the AVPV, PVN, SON, SCN, and CP were obtained, according to the rat brain atlas of Paxinos and Watson [67]. The brain sections were mounted on polyethylene naphthalate membrane slides (Leica), fixed with ice-cold 5% acetic acid in ethanol for 3 min, stained with ice-cold 0.2% cresyl fast violet for 30 seconds, rinsed in ice-cold RNase free water for 1 minute two times, and dried with cool air. The nuclei were then dissected out from cresyl fast violet-stained brain sections by the LMD system and collected in a tube containing 70 µL RNA extraction buffer from the RNeasy Micro Kit (Qiagen, Valencia, CA, USA).

Total RNA was extracted and purified using the RNeasy Micro Kit (Qiagen), according to the manufacturer’s protocol. First-strand cDNA was synthesized using the PrimeScript RT Master Mix (TaKaRa Bio, Otsu, Japan). Standard samples for *Esr2* and glyceraldehyde-3-phosphate dehydrogenase (*Gapdh*), which is a housekeeping gene, were prepared by mixing an equal volume of unknown cDNA sample and serially diluting them in an EASY Dilution solution (TaKaRa Bio). Real-time PCR was performed using a Thermal Cycler Dice Real-Time System (TaKaRa Bio). A 2-µL aliquot of standard or unknown sample was amplified in a 25-µL reaction mixture containing 400 nM of each gene-specific primer (*Esr2* forward primer: TACCCACACTGCACTTCCCAG, *Esr2* reverse primer: CCAAGTGGGCAAGGAGACAGA; *Gapdh* forward primer: AGTCTACTGGCGTCTTCACCAC, *Gapdh* reverse primer: TGGTTCACACCCATCACAAACATG) and 12.5 µL of 2× SYBR Premix Ex Taq (TaKaRa Bio). The amount of *Esr2 m*RNA was normalized by dividing the amount of *Gapdh* mRNA for each sample.

#### 4.3.5. RNAscope ISH and Immunofluorescence

We performed RNAscope ISH to detect *Esr2* mRNA as described previously, without counter staining the nuclei with hematoxylin. Following the reaction with Fast Red in RNAscope, samples were immediately processed for immunofluorescence, as described below. Slides were washed in PBS containing 1% (*v*/*v*) Triton X-100 (PBST) and were incubated in blocking buffer consisting of 5% bovine serum albumin for 60 min at RT. Consequently, the samples were incubated with a primary antibody of each neuropeptide diluted 1:1000 in PBST at 4 °C overnight. The following day, after washing with PBS, the samples were incubated with a solution containing secondary antibody diluted 1:200 in PBS at RT for 2 h. All details of primary and secondary antibodies used in this study are shown in Table 1. After the reaction was stopped by washing in PB, the tissue was cover-slipped using CC/mount medium (Cosmo Bio, Tokyo, Japan).

#### 4.3.6. Images and Data Analysis

The left side of the brain section was analyzed. The signals for *Esr2* by RNAscope were readily apparent as red particles, visible under a light microscope, and under fluorescent microscopy using a Texas Red (531–635 nm) filter. Images of each section used in single RNAscope ISH were acquired using a BX51 microscope fitted with a DP73 digital camera (Olympus, Tokyo, Japan). Confocal images were captured using a confocal laser scanning microscope FV10i (Olympus, Tokyo, Japan) for the analysis of the co-expression between *Esr2* and each neuropeptide.

The numbers of neuropeptide-immunoreactive (ir) cells and the co-expression between *Esr2* and each neuropeptide-ir cell was manually counted. In this experiment, neuropeptide-ir neurons containing three or more particles of *Esr2* were considered to be “co-expressed” cells (Appendix A), according to preceding studies [68,69,70]. We also assessed the numbers of cells with more than five particles as a “relatively high-expression cell” and calculated percentages of these cells in all *Esr2*-expressing cells to roughly estimate the expression level on a cellular level. The PVN is a phenotypically and functionally heterogeneous nucleus and distribution patterns of neuronal types vary along the rostro-caudal axis in the PVN [71]. Thus, we analyzed at different rostro-caudal levels of the PVN (−0.9 to −1.32 mm, −1.44 to −1.56 mm, −1.72 to −1.92 mm, and −2.04 to −2.16 mm from bregma) as previous studies [14,32]. The numbers of brain sections used for counting were 14–15 sections for *Esr2*/kisspeptin in the AVPV, 22–24 sections for *Esr2*/kisspeptin in the ARC, 14–15 sections for *Esr2*/AVP in the PVN, 12–13 sections for *Esr2*/OXT in the PVN, 10–13 sections for *Esr2*/CRF in the PVN, 15 sections for *Esr2*/AVP in the SON, and 13–15 sections for *Esr2*/OXT in the SON. The percentage of neuropeptide-ir cells expressing *Esr2* mRNA was calculated by dividing the number of co-expressed cells by the total number of respective neuropeptide-ir cells and multiplying it by 100. The observer was unaware of the experimental groups.

#### 4.3.7. Statistical Analysis

Two-way ANOVA was used for the analysis of co-expression of *Esr2* in each neuropeptide-ir cells in the PVN, and, hence, to assess the effect of the rostro-caudal level and sex, and the interaction of these factors. We used the test of simple main effects when the effects of the interaction between the main factors were significant in the two-way ANOVA. Post-hoc analysis with a Bonferroni correction was performed to compare the co-expression ratio among different rostro-caudal levels. Apart from the PVN, the Student’s *t*-test was conducted to compare the sex difference. We did not statistically analyze the *Esr2* mRNA levels measured by qPCR because of the low number of animals used. IBM SPSS Statistics v. 20.0 (IBM, Armonk, NY, USA) was used for data analysis. A value of *p* < 0.05 was considered statistically significant.

## 5. Conclusions

In this study, we determined the neurochemical characteristics of *Esr2*-expressing neurons in the rat hypothalamus using RNAscope ISH combined with immunofluorescence. The majority of kisspeptin neurons in female AVPV expressed *Esr2*, whereas a low number of kisspeptin neurons in the ARC co-expressed *Esr2*. In the PVN, we observed a high ratio of *Esr2* co-expression in AVP neurons and a low ratio, respectively, in OXT and CRF neurons. We believe our findings have the capacity to provide a greater understanding regarding estrogen signaling via ERβ.

## Figures and Tables

**Figure 1 ijms-21-00115-f001:**
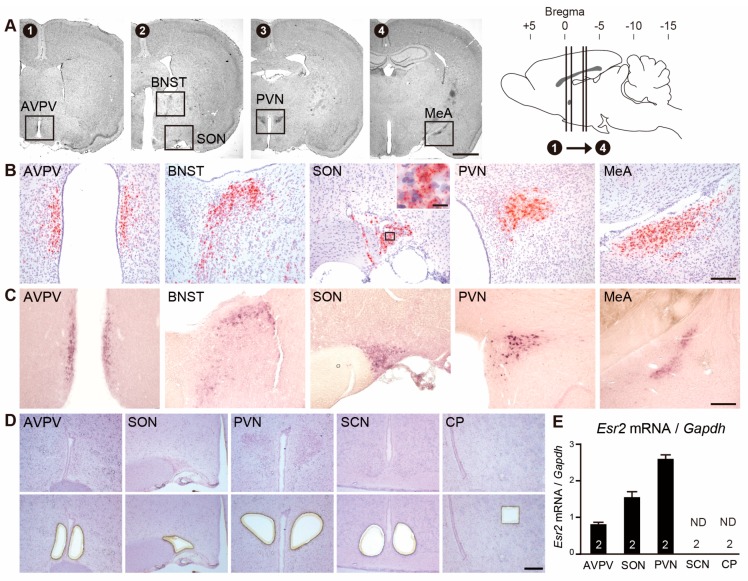
Comparison of *Esr2* signals by RNAscope with *Esr2* signals by conventional *in situ* hybridization (ISH) and expression levels of *Esr2* mRNA by qPCR. (**A**) Nissl-stained images (1–4) and schematic sagittal brain section demonstrate the rostro-caudal levels of the rat brain. Insets in the Nissl-stained images show anatomical loci of the AVPV, BNST, SON, PVN, and MeA. Representative photomicrographs of *Esr2* mRNA signals by RNAscope (**B**) and by conventional ISH (**C**) in the AVPV, BNST, SON, PVN, and MeA. (**D**) Representative photographs exhibiting brain sections before (upper panels) and after (lower panels) isolation by laser-microdissection. Scale bars indicate 2 mm in all panels in A, and 200 µm in all panels in B, C, and D. Inset in the SON panel in B shows higher magnification of cells containing *Esr2* signals (scale bar = 20 µm). (**E**) Quantification of the *Esr2* mRNA in the micro-dissected samples by qPCR. Numbers in each column represent the number of animals used. Values are depicted as the mean ± standard error of the mean (SEM). ND indicates “not-detected”. AVPV: anteroventral periventricular nucleus; BNST: bed nucleus stria terminalis; SON: supraoptic nucleus; PVN: paraventricular nucleus of the hypothalamus; MeA: medial amygdala; SCN: suprachiasmatic nucleus; CP: corpus striatum.

**Figure 2 ijms-21-00115-f002:**
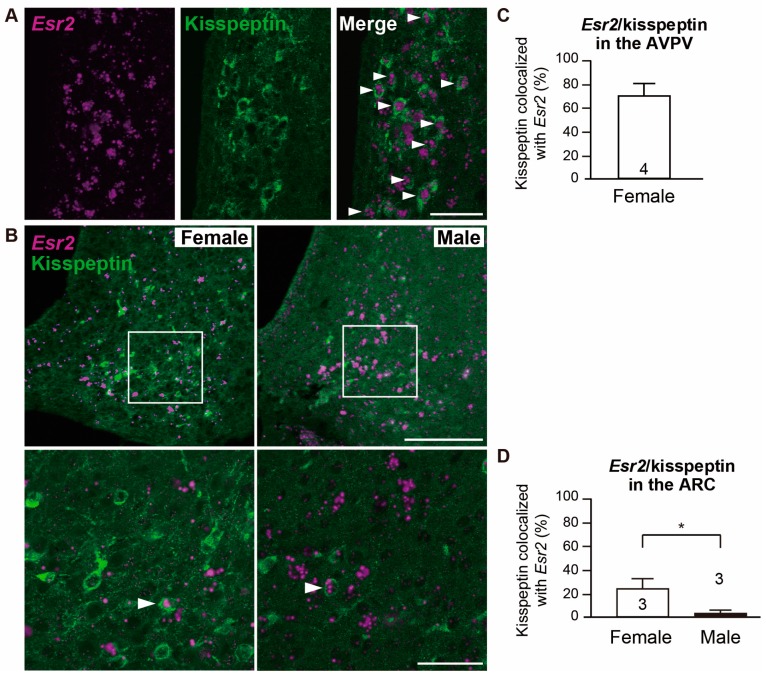
Co-expression of *Esr2* and kisspeptin in the anteroventral periventricular nucleus (AVPV) and arcuate nucleus (ARC). Representative photomicrographs show dual labeling of *Esr2* mRNA (magenta) and kisspeptin (green) in the AVPV of female rat (**A**) and ARC of both sexes (**B**). High magnification images (the bottom lower images) correspond to the boxed areas in the overviews. Arrowheads indicate double-labeled cells. Scale bars measure 50 µm in A, and 200 µm in the low magnification, and 50 µm in the high magnification in B. The percent of kisspeptin-ir cells co-expressed with *Esr2* mRNA in the AVPV (**C**) and ARC (**D**). Values are depicted as the mean ± SEM. Numbers in each column represent the number of animals used. * *p* < 0.05.

**Figure 3 ijms-21-00115-f003:**
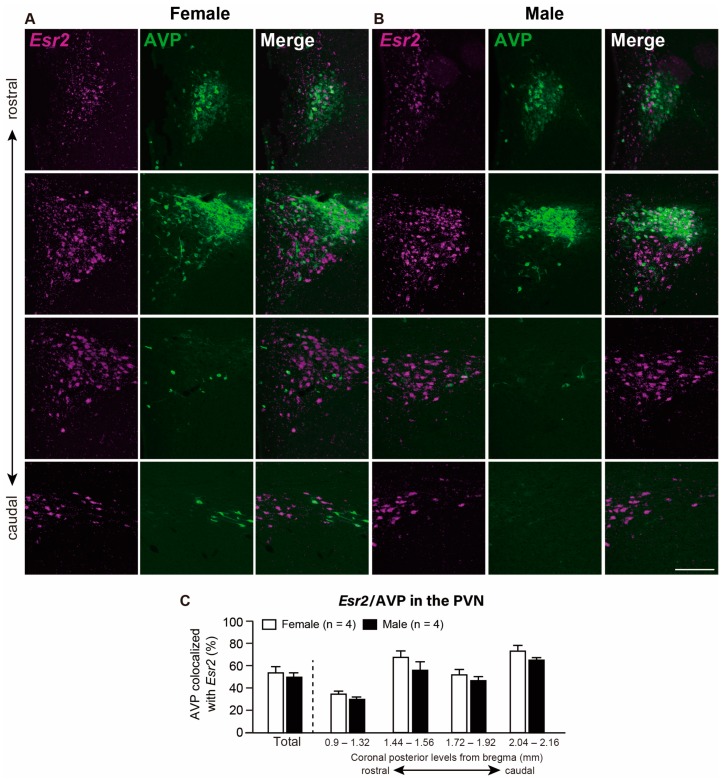
Co-expression of *Esr2* and arginine vasopressin (AVP) in the paraventricular nucleus of the hypothalamus (PVN). Representative photomicrographs show dual labeling of *Esr2* mRNA (magenta) and AVP (green) in the PVN of female (**A**) and male (**B**) rats. Scale bars indicate 200 µm in A and B. (**C**) The percentage of AVP-ir cells co-expressed with *Esr2* mRNA in the whole PVN and the PVN at different rostro-caudal levels (from bregma −0.9 to −2.16 mm). Values represent the mean ± SEM.

**Figure 4 ijms-21-00115-f004:**
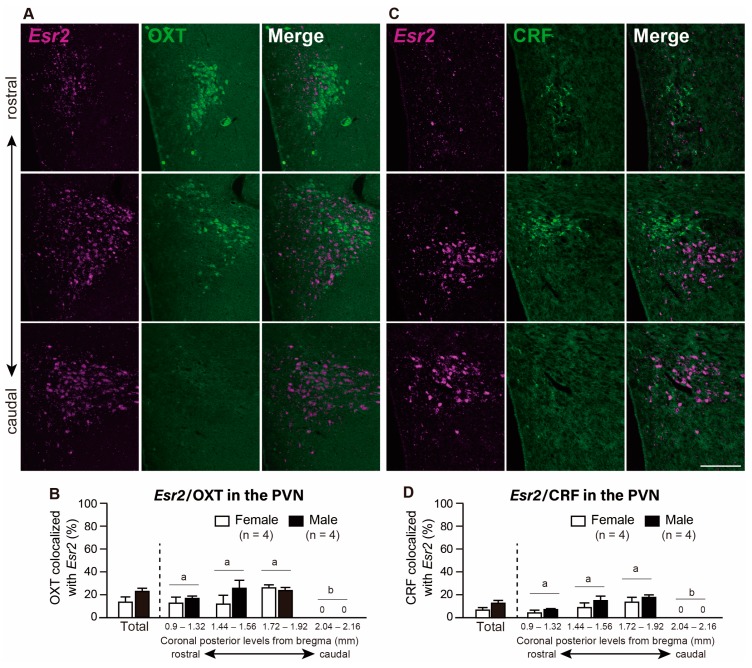
Co-expression of *Esr2*/oxytocin (OXT) and *Esr2*/corticotropin-releasing factor (CRF) in the paraventricular nucleus of the hypothalamus (PVN). Representative photomicrographs show dual labeling of *Esr2* mRNA (magenta) and OXT (green) in the PVN of female rats (**A**), and dual labeling of *Esr2* mRNA (magenta) and CRF (green) in the PVN of female rats (**C**). Arrowheads indicate double-labeled cells. Scale bar indicate 200 µm in A and C. The bar graph shows the percentage of co-expression of *Esr2* mRNA/OXT-ir cells (**B**) and *Esr2* mRNA/CRF-ir cells (**D**) in the whole PVN and the PVN at different rostro-caudal levels (from bregma −0.9 to −2.16 mm). Values represent the mean ± SEM. Differences in values that are significantly (*p* < 0.05) different from each other are indicated by letters * *p* < 0.05.

**Figure 5 ijms-21-00115-f005:**
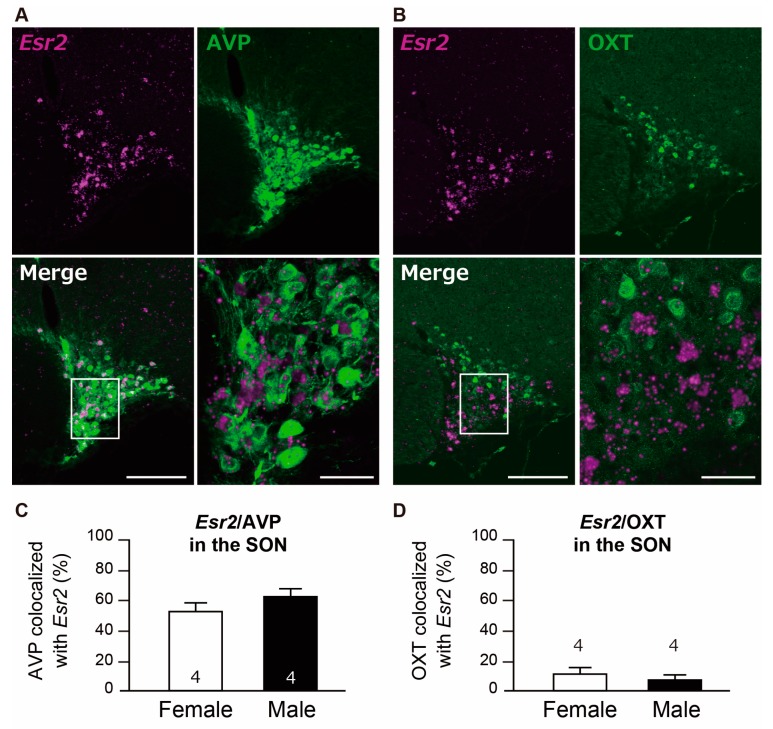
Co-expression of *Esr2*/arginine vasopressin (AVP) and *Esr2*/oxytocin (OXT) in the supraoptic nucleus (SON). Representative photomicrographs show dual labeling of *Esr2* mRNA (magenta) and AVP (green) in the SON of female rats (**A**), and dual labeling of *Esr2* mRNA (magenta) and OXT (green) in the SON of female rats (**B**). High magnification images (the bottom right images) correspond to the boxed areas in the overviews in A and B. Scale bar measures 50 µm for high magnifications and 200 µm for low magnification images in A and B. The bar graph shows the co-expression percentage of *Esr2* mRNA/AVP-ir cells in the SON (**C**) and *Esr2* mRNA/OXT-ir cells in the SON (**D**). Values represent the mean ± SEM. The number in each column indicates the number of animals used.

**Table 1 ijms-21-00115-t001:** Primary and secondary antibodies used.

Target	Primary Antibody	Manufacturer, Catalog # or Source	Species Raised in Monoclonal or Polyclonal	Secondary Antibody	Manufacturer, Catalog # or Source
Kisspeptin	Mouse monoclonal kisspeptin antibody	Takeda Pharmaceutical Co., Ltd.	Mouse; monoclonal	Donkey Alexa 488 anti-mouse	Abcam, ab150105
Arginine-vasopressin	Guinea Pig anti-(Arg8)-Vasopressin	Peninsula, T5048	Guinea pig; polyclonal	Donkey Dy Light 488 anti-guinea pig	Jackson, 706-485-148
Oxytocin	Anti-Oxytocin Antibody	Merck Millipore, AB911	Rabbit; polyclonal	Donkey Alexa 488 anti-rabbit	Abcam, ab150073
Corticotropin-releasing factor	Rabbit anti- Corticotropin-releasing factor	Peninsula, T4037	Rabbit; polyclonal	Donkey Alexa 488 anti-rabbit	Abcam, ab150073

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
