# Peer review of "Neurochemical Characterization of Neurons Expressing Estrogen Receptor β in the Hypothalamic Nuclei of Rats Using in Situ Hybridization and Immunofluorescence"

_ijms, 2019, doi:10.3390/ijms21010115_

Round 1

Reviewer 1 Report

This manuscript examines the distribution and neuronal phenotype of Esr2 expressing neurons in the hypothalamus using RNAscope technology.   This is a novel approach for examining Esr2 neurons and helps the field which has struggled because of lack of usable reagents.  The authors have very nicely validated the approach and have importantly examined the distribution of Esr2 mRNA in various neuropeptide neurons.  Importantly, results from kisspeptin neurons is novel and will help move the field forward.  Much of the data from the PVN and SON neuropeptide neurons have been reported previously using other approaches and although not necessarily new information, it is nice to see that much of it replicates prior studies that used other approaches.  Nonetheless, there are a few concerns, listed below, which should be addressed by the authors.

Pg. 12,   identify the rat brain atlas used.

Pg 13:   coexpression was based on neuropeptide neurons “….containing 3 or more particles of EsR2…….”.     What is the significance of 3 particles and what was the rationale for  setting that value as the threshold? 

Pg 13:    It appears that successive sections were examined for cell counts.  Was there a procedure in place to avoid double counting errors?

One of the advantages of RNAscope is the ability to quantify levels of transcript by grain counting approaches.  However, it appears that the authors have not taken advantage of this by only showing quantification of microdissected PVN and not levels of Esr2 mRNA in individual cell types.   Could the apparent lack of expression in some cell phenotypes be due to low levels of expression instead?

Perhaps a very short discussion of the differences in co-localization between ERbeta and other PVN neuropeptides between mouse and rat should be included as they seem to be somewhat different.

Although the discussion headings list the SON, there is very little discussion of the results in the SON and the discussion by far is centered on the PVN.   The results from SON should be addressed in more detail and prior studies showing colocalization or not in SON neuropeptide neurons should be included.   Moreover, the possibility is that a major component of the circulating levels of these hormones is actually derived from the SON and this should also be included.   

There is little discussion of where the neuropeptide neurons in hypothalamus that express Esr2 project to, particularly those in the PVN.  

The RNAscope approach used sections cut at a thickness of 16 um whereas the ISH with the DIG-RNA probes used 40um thick sections.  It is difficult to compare the results from both the 16 and 40 um thick sections, particularly when the depth that each probe penetrates tissue is unknown as are differences between the two types of probes in terms of length and labelling.   This should be addressed.

Although the authors have determined the % of neuropeptide neurons that express Esr2 mRNA, it would also be informative if the number of Esr2 mRNA cells that express each neuropeptide was also reported and whether there are sex differences in these ratios.

Although the authors have discussed the regulation of kisspeptin by estrogens, the sex differences that they report are in intact male and female rats, which begs the question of what impact androgens (predominantly in males) may impact the results.

Author Response

We sincerely appreciate your valuable comments and constructive suggestions on our manuscript. In accordance with your comments and suggestions, we have revised the manuscript.

Comments and Suggestions for Authors

This manuscript examines the distribution and neuronal phenotype of Esr2 expressing neurons in the hypothalamus using RNAscope technology. This is a novel approach for examining Esr2 neurons and helps the field which has struggled because of lack of usable reagents. The authors have very nicely validated the approach and have importantly examined the distribution of Esr2 mRNA in various neuropeptide neurons. Importantly, results from kisspeptin neurons is novel and will help move the field forward. Much of the data from the PVN and SON neuropeptide neurons have been reported previously using other approaches and although not necessarily new information, it is nice to see that much of it replicates prior studies that used other approaches. Nonetheless, there are a few concerns, listed below, which should be addressed by the authors.

Response:

We would like to thank that you described positive comments.

Comment 1:

Pg. 12, identify the rat brain atlas used.

Response 1:

We added the reference (see Line 470).

Comment 2:

Pg 13: coexpression was based on neuropeptide neurons “….containing 3 or more particles of EsR2…….”. What is the significance of 3 particles and what was the rationale for setting that value as the threshold?

Response 2:

The manufacture producing RNAscope recommend to set the criteria depending on the expression level of target genes [see https://acdbio.com/technical-support/solutions, Scoring Guidelines]. In setting the criteria, we have searched many literatures using RNAscope ISH for co-expression analysis. Although there was no specific de facto standard value for threshold, many researchers used values ranging from 1 to 10 grains; Samineni et al. used 5 particles/cell as a critera [Samineni et al., Nature Communication, 2019; 10(1): 4356.], Steinkellne et al. used 4 particles [Steinkellne et al., Journal of Clinical Investigation, 2018; 128(2): 774–788], and Lanfranco et al. used 1 particle [Lanfranco et al., Frontiers in Neuroanatomy, 2017; 11: 137]. Therefore, we think the threshold used in this study is reasonable value.

We added these references in the Materials and Methods (see Line 512).

Comment 3:

Pg 13: It appears that successive sections were examined for cell counts. Was there a procedure in place to avoid double counting errors?

Response 3:

In this experiment, four series of brain coronal sections were cut using a cryostat. Each series of sections was used for a combination of RNAscope and immunofluorescence. So, we did not use adjacent successive sections but used every fourth section from a brain in each staining condition. So, sections in each staining condition can be distinguished from each other. Additionally, to avoid double counting, we carefully checked high-magnification images used in cell counting by comparing with corresponding low-magnification images.

We apologize for our lack of the explanation in the original manuscript. We rewrote the sentences as follows: (see Line 394–401).

Four series of brain coronal sections were cut at a thickness of 16 μm for single RNAscope ISH and for a combination of RNAscope and immunofluorescence, and cut at a thickness of 40 μm for conventional ISH with DIG-RNA probe using a cryostat (CM3050; Leica, Wetzlar, Germany). We used 40 μm-thick sections for DIG-ISH in accordance with our previous study [54], in which we optimized the ISH procedure for detection of low-expression targets. Each series of sections was stored separately in cryoprotectant at 25 and then used for single RNAscope ISH, a combination of RNAscope and immunofluorescence, and conventional ISH.

Comment 4:

One of the advantages of RNAscope is the ability to quantify levels of transcript by grain counting approaches. However, it appears that the authors have not taken advantage of this by only showing quantification of microdissected PVN and not levels of Esr2 mRNA in individual cell types. Could the apparent lack of expression in some cell phenotypes be due to low levels of expression instead?

Response 4:

As you pointed out, applicability for semi-quantification of genes in single cell is one of the major advantages of RNAscope. According to your advice, we assessed the numbers of cells with ≧5 particles as “relatively high-expression cell” and calculate percentages of these cells in all Esr2-expressing cells (≧3 particle/cell). We added descriptions about this data in Materials and Methods, Results, and Discussion, and show summarized graph in a new supplemental figure (see Supplemental figure 3). We added the sentence as follows:

In Results,

“The composition of cells which have more than 5 particles/cell within Esr2-expressing cells was 87% AVPV kisspeptin, and 45% in ARC kisspeptin of females (Supplemental Fig. 3A). Note that we did not perform analysis in male AVPV because males have few kisspeptin neurons in the AVPV.” (see Line 117–121)

“Within these Esr2-expressing AVP neurons in the PVN, 74% and 80% of neurons showed 5 signal particles/cell in female and male, respectively (Supplemental Fig. 3B).” (see Line 137–139)

“Within these Esr2-expressing OXT neurons, 55% and 51% of neurons showed ≧5 signal particles/cell in female and male, respectively.” (see Line 152–153)

“Additionally, majority of Esr2-expressing AVP neurons in the SON showed ≧5 particles/cell (85% in females and 79% in males), whereas the percentage of cells with ≧5 particles/cell in Esr2-expressing OXT neurons was low (36% in females and 37% in males).” (see Line 173–175)

In Discussion,

Moreover, the percentage of cells with 5 Esr2 signals was relatively higher in the AVP neurons compared with OXT neurons in the PVN and SON.” (see Line 249–250)

In Materials and Methods,

In this experiment, neuropeptide-ir neurons containing 3 or more particles of Esr2 were considered “co-expressedcells (Supplemental Fig. 2I), according to preceding studies [Samineni et al., Nature Communication, 2019; 10(1): 4356; Steinkellne et al., Journal of Clinical Investigation, 2018; 128(2): 774–788; Lanfranco et al., Frontiers in Neuroanatomy, 2017; 11: 137]. We also assessed the numbers of cells with more than 5 particles as “relatively high-expression cell”, and calculated percentages of these cells in all Esr2-expressing cells to roughly estimate expression level in cellular level.” (see Line 511–515)

However, please kindly note the following points:

According to manufacturer’s recommendation, optimization of staining condition for semi-quantification prior to assessment is desirable (please see https://acdbio.com/technical-support/solutions). Since we did not perform this optimization procedure for quantification, semi-quantification with recommended scaling index is not applicable. Hence, we performed cell counting with ≧5 particles as a suboptimal choice.

Comment 5:

Perhaps a very short discussion of the differences in co-localization between ERbeta and other PVN neuropeptides between mouse and rat should be included as they seem to be somewhat different

Response 5:

Thank you for your suggestion. Regarding the PVN of mice, Oyola and his colleague have observed many fewer ERβ/AVP co-expressing cells in the mice PVN, and there was sex difference, with females having more co-localized cells (45%) than males (11%) in the caudal PVN [Oyola et al, Journal of Comparative Neurology, 2007; 525(17): 3666-3682]. This group also reported that the percentage of OXT neurons co-expressing ERβ is more than 80% through the PVN of both sexes. These findings suggest there exists a species difference between rat and mice.

We added the following sentences in Discussion (see Line 260–269).

Regarding the PVN of mice, Oyola et al. reported a relatively low ERβ/AVP co-expression with sex difference (45% in females vs. 11% in males in the caudal PVN) [Oyola et al., Journal of Comparative Neurology, 2017; 525, 3666-3682; Biag et al., Journal of Comparative Neurology , 2012; 520, 6-33] using a ERβEGFP transgenic mice model. This group also reported a high co-expression of ERβ in OXT neurons (>80% in both sexes) in the PVN. These findings suggest there exists a species difference in neurochemical characteristics of Esr2-expressing neurons between rats and mice.

Comment 6:

Although the discussion headings list the SON, there is very little discussion of the results in the SON and the discussion by far is centered on the PVN. The results from SON should be addressed in more detail and prior studies showing colocalization or not in SON neuropeptide neurons should be included. Moreover, the possibility is that a major component of the circulating levels of these hormones is actually derived from the SON and this should also be included.

Response 6:

As you suggested, we added the references regarding the co-localization of AVP and OXT neurons with ERβ in the SON and regarding the involvement of the SON in neurosecretory system such as water balance as follows:

In the SON, our present result is consistent with previous studies, in which the co-localization of AVP neurons with ERβ-ir cells is high (72% [Suzuki and Hand, Journal of Comparative Neurology, 2005; 484, 28–42] and 88% [Hrabovszky et al., Journal of Comparative Neurology, 2004; 473, 315–33])” (see Line 269–271)

 “A well-documented physiological function of AVP neurons in both the PVN and SON is the maintenance of water balance which is influenced by estrogens [39, 40]. The neurons responsible for the maintenance of water homeostasis are the magnocellular neurosecretory cells located in the PVN and SON that projects its axons toward the posterior pituitary and is subsequently released into the circulation.” (see Line 274–278)

Also, in the SON, we observed that the co-localization of the OXT neurons with ERβ-ir cells is relatively lower than the AVP neurons as a previous study [Hrabovszky et al., Journal of Comparative Neurology, 2004; 473, 315–33])” (see Line 318­–319)

Comment 7:

There is little discussion of where the neuropeptide neurons in hypothalamus that express Esr2 project to, particularly those in the PVN.

Response 7:

Your points are well-taken. We added the short discussion about the role of neuropeptides in the PVN projecting to other brain regions apart from pituitary gland as follows:

Regarding the AVP neurons in the PVN (see Line 292–296):

“Additionally, a recent study disclosed that galanin-expressing neurons in the medial preoptic area directly receive the inputs from the AVP neurons in the PVN and SON without a clear sex difference [Kohl et al., Nature, 2018; 556(7701): 326–331]. The galanin-expressing neurons in the medial preoptic area are associated with parental behaviors, and therefore the estrogen action via ERβ on AVP neurons in the PVN partially results in an effect on the mechanism of parental behaviors in both sexes.

Regarding the OXT in the PVN (see Line 330–338):

In addition to its neuroendocrine function, OXT neurons in the PVN play roles in modulating wide variety of brain functions through the direct axonal projections to multiple brain areas such as cortex and central amygdala. The former neuronal pathway seems to be implicated in maternal behavior [Yuan and Hou, Frontiers in Behavioral Neuroscience, 2015; 9: 311], and the latter one is related to fear response [Knobloch et al., Neuron, 2012; 73(3): 553–566]. In addition, parvocellular neurons in the PVN project to sympathetic and parasympathetic centers in the brainstem and spinal cord [Swanson and Kuypers , Journal of Comparative Neurology, 1980; 194(3): 555–570]. It has been proposed the involvement of PVN OXT neurons in the modulation of sympathetic and parasympathetic outflows to spinal cord[Chen et al., Neuroscience, 2003; 118(3): 797–807]. Further analysis of ERβ expression in each OXT neuronal populations depending on its projection site is necessary to fully elucidate the ERβ functions.”

Comment 8:

The RNAscope approach used sections cut at a thickness of 16 um whereas the ISH with the DIG-RNA probes used 40um thick sections. It is difficult to compare the results from both the 16 and 40 um thick sections, particularly when the depth that each probe penetrates tissue is unknown as are differences between the two types of probes in terms of length and labelling. This should be addressed.

Response 8:

We totally agree with your comment regarding the difficulty in comparing the results from two different methodologies. However, the point we want to put weight on in Discussion is that RNAscope provides qualitatively better sensitivity in detection of Esr2 compared with conventional DIG-ISH. We would appreciate your understanding that we did not intend to quantitatively compare the results from these two methodologies.

In previous studies, we have optimized DIG-ISH procedure to detect low-expression target genes [Higo et al., Journal of Neuroendocrinology, 2016; 28(4). doi: 10.1111/jne.12356]. In accordance with the study, we used 40 μm-thick sections for DIG-ISH in this study. Thickness of the sections used in RNAscope ISH (16 μm) was also empirically set by optimization in preliminary trials.

Since our original manuscript may mislead reviewer and reader as if we intended quantitative comparison of results obtained by two methodologies, we rewrote and added the description about this matter as follows:

In Results (see Line 79–81):

Likewise, conventional ISH using DIG-RNA probes revealed Esr2 mRNA positive cells in these nuclei (Fig. 1C), which were relatively obscurer signals compared to the ones detected by RNAscope.

In Discussion (see Line 196–199):

In addition to the strong signals in the nuclei mentioned above, we observed moderate RNAscope signals in several brain regions including OVLT and VMHvl, as opposed to the weak obscure or no signals detected by conventional ISH in these regions, suggesting that sensitivity of RNAscope for Esr2 was qualitatively better than that of conventional ISH.

In Materials and Methods (see Line 397–399):

We used 40 μm-thick sections for DIG-ISH in accordance with our previous study [54], in which we optimized the ISH procedure for detection of low-expression targets [Higo et al., Journal of Neuroendocrinology, 2016; 28(4). doi: 10.1111/jne.12356].

Comment 9:

Although the authors have determined the % of neuropeptide neurons that express Esr2 mRNA, it would also be informative if the number of Esr2 mRNA cells that express each neuropeptide was also reported and whether there are sex differences in these ratios.

Response 9:

We agree with your comment that the percentage of Esr2-expressing cells that co-express each neuropeptide would be informative. However, we cannot provide accurate percentage because of methodological difficulties described below.

In calculating the percentage of neuropeptide positive neurons that express Esr2 mRNA, we can assess the total number of neuropeptide cells based on the immunoreactive signals against particular neuropeptide. However, it is difficult to count cells based solely on RNAscope signals in brain sections, since the cell boundaries cannot be distinguished in RNAscope stained images because of the particulate feature of signals. This difficulty is prominent especially in detection of low-expression target in thick sections. To assess the total number of Esr2-expessing neurons, it is necessary to perform RNAscope in combination with immunostaining against cell surface or cytoskeleton markers. However, we did not have enough samples for additional staining.

We recognize this imperfection in fundamental data of Esr2-expressing neurons is a weakness of the current study, and we are just trying to fulfill this lack of information in the future study.

Comment 10:

Although the authors have discussed the regulation of kisspeptin by estrogens, the sex differences that they report are in intact male and female rats, which begs the question of what impact androgens (predominantly in males) may impact the results.

Response 10:

Thank you for pointing out the lack of discussion about androgen effects in our original manuscript. Actually, kisspeptin expression is regulated by androgens as well as estrogens. Many previous reports have shown that the expression level of kisspeptin have clear sex-difference, with female having higher expression than males in both the AVPV and ARC, as we showed in this study. Previous researches suggest that the low expression level of kisspeptin in male results from persistent suppression by perinatal androgen surge and circulating androgen in adulthood. We presume that the sex-difference in Esr2-expressing kisspeptin neurons in the ARC may reflect the sex difference in the total number of kisspeptin neurons as a consequence of different sex steroid condition. We rewrote the manuscript as follows (see Line 237–245):

 “In this study, kisspeptin expression showed clear sex-difference with lower expression in males than females in the ARC, which is consistent with our previous study [28]. It is suggested that not only estrogen but also testosterone play a role in this sexual dimorphism in kisspeptin; perinatal testosterone surge persistently suppress kisspeptin after aromatization to estradiol in the brain [29–31], and high circulating testosterone also suppress kisspeptin expression in adulthood [30]. Co-expression analysis revealed distinct sex differences in Esr2 co-expression in kisspeptin neurons in the ARC. This result may reflect the sex difference in the total number of kisspeptin cells in the ARC (180.0 ± 29.0 cells in females vs. 41.5 ± 6.20 cells in males), as a consequence of the difference in intrinsic sex steroid milieu between sexes.

We tried to respond to your suggestion as much as we can and we think that our manuscript was improved. We hope that the revisions in the manuscript and our responses will be sufficient to make our manuscript suitable for publication. I hope you are satisfied with revised manuscript. We would like to thank you all for your kind cooperation.

Sincerely yours,

Shimpei Higo, PhD

Corresponding author of Manuscript ID: ijms-644006

Reviewer 2 Report

This paper addresses the detailed data regarding the neurochemical characteristics of neurons expressing Esr2 mRNA in the hypothalamus of male and female rats using RNAscope in situ hybridization (ISH) for Esr2 mRNA combined with immunofluorescence against 4 neuropeptides: kisspeptin, AVP, OXT, and CRF. Actually, they carefully confirmed the availability of RNAscope ISH for Esr2 mRNA, even though the numbers of animals are fairly limited. They also found some signals for Esr2 mRNA by RNAscope method in the organum vasculosum laminae terminalis, the ventrolateral hypothalamic tract, and in the ventrolateral parts of ventromedial hypothalamus, in which regions the signals were not detected obviously by using the digoxigenin-labeled RNA probe. The experimentations seem to be accurately performed and the results of their experiments are useful for the research on estrogen signaling. This paper would be complete after a minor revision for the point listed below.

 [a minor point]

Several ERβ isoforms in brain are reported in many literatures; however, the authors never discussed this matter and they did not mention whether which isoforms or all isoforms can be detected by their RNAscope probe used in the experiments. It would be better to add some information on the isoform specificity of their RNAscope probe to their manuscript. This information might lead to some clue for better understanding of the difference in co-localization ratio of Esr2 mRNA and brain neuropeptides in the future research.

Author Response

We hope to express our appreciation for your comments on our manuscript. We think that the comments have helped us improve the paper.

Comments and Suggestions for Authors

This paper addresses the detailed data regarding the neurochemical characteristics of neurons expressing Esr2 mRNA in the hypothalamus of male and female rats using RNAscope in situ hybridization (ISH) for Esr2 mRNA combined with immunofluorescence against 4 neuropeptides: kisspeptin, AVP, OXT, and CRF. Actually, they carefully confirmed the availability of RNAscope ISH for Esr2 mRNA, even though the numbers of animals are fairly limited. They also found some signals for Esr2 mRNA by RNAscope method in the organum vasculosum laminae terminalis, the ventrolateral hypothalamic tract, and in the ventrolateral parts of ventromedial hypothalamus, in which regions the signals were not detected obviously by using the digoxigenin-labeled RNA probe. The experimentations seem to be accurately performed and the results of their experiments are useful for the research on estrogen signaling. This paper would be complete after a minor revision for the point listed below.

Response:

We would like to thank your careful reading and favorable comments.

Comment:

 [a minor point]

Several ERβ isoforms in brain are reported in many literatures; however, the authors never discussed this matter and they did not mention whether which isoforms or all isoforms can be detected by their RNAscope probe used in the experiments. It would be better to add some information on the isoform specificity of their RNAscope probe to their manuscript. This information might lead to some clue for better understanding of the difference in co-localization ratio of Esr2 mRNA and brain neuropeptides in the future research.

Response:

Thank you for suggesting a new viewpoint. Both the conventional DIG-labeled probe and RNAscope probe used in this study were designed based on the full-length Esr2 gene (GenBank NM_012754.1). Considering the target sequences of the probes, both probes can detect several isoforms such as ER-β2 variant (54 bp insertion between exons 5 and 6), and δ3 variants (deletion of the 3rd exon). Because of this pan-specificity of the probes to multiple ERβ isoforms, we cannot determine the isoform expression profile among stained neurons. Further studies using isoform-specific probes is needed to provide the isoform profiles.

We added a paragraph in Discussion as follows, (see Line 201–212)

“Several splice variants of ERβ mRNA have been reported in the brain [Weiser, et al. Brain Research Reviews, 2008; 57(2): 309–320] such as ER-β2 variant (54bp insertion between exons 5 and 6), and δ3 variants (deletion of the 3rd exon). It is suggested that these isoforms play roles in differential regulation of estrogen among brain regions [Handa, et al. Journal of Neuroendocrinology, 2012; 24(1): 160–173] as a result of the functional diversity of these isoforms in many properties including the transcriptional activation and ligand-binding affinity. In this study, both the DIG-labeled probe and RNAscope probe can detect these variants as well as wild-type Esr2, while it is difficult to determine the expression profile of these isoforms using these probes because of its pan-specificity to multiple isoforms. Further characterization of the isoform profile in each Esr2-expressing neuronal population is needed to better understand the role ERβ functions in the multiple brain areas.”

We hope that the revisions in the manuscript and our responses will be sufficient to make our manuscript suitable for publication. We would like to thank you all for your kind cooperation.

Sincerely yours,

Shimpei Higo, PhD

Corresponding author of Manuscript ID: ijms-644006

Round 2

Reviewer 1 Report

The authors have satisfactorily addressed all of my concerns.